# Complexation: An Interesting Pathway for Combining Two APIs at the Solid State

**DOI:** 10.3390/pharmaceutics14091960

**Published:** 2022-09-16

**Authors:** Fucheng Leng, Oleksii Shemchuk, Koen Robeyns, Tom Leyssens

**Affiliations:** Institute of Condensed Matter and Nanosciences, Université Catholique de Louvain, 1 Place Louis Pasteur, B-1348 Louvain-La-Neuve, Belgium

**Keywords:** solid state, crystal engineering, drug–drug complexes, solubility improvement

## Abstract

Combining different drugs into a single crystal form is one of the current challenges in crystal engineering, with the number of reported multi-drug solid forms remaining limited. This paper builds upon an efficient approach to combining Active Pharmaceutical Ingredients (APIs) containing carboxylic groups in their structure with APIs containing pyridine moieties. By transforming the former into their zinc salts, they can be successfully combined with the pyridine-containing APIs. This work highlights the successfulness of this approach, as well as the improvement in the physical properties of the obtained solid forms.

## 1. Introduction

The last few decades have witnessed a rapid increase in life expectancy around the world. This is, however, accompanied by an increased number of cases of age-associated chronic and complex diseases such as diabetes, cancer, and cardiovascular disorders [1,2]. Multi-drug combinations have demonstrated superiority compared to traditional mono-therapy approaches for the treatment of such diseases [3,4,5,6]. Administrating several drugs in parallel will decrease the patient’s compliance, especially for elderly people. Formulating different drugs into one solid dosage form (fixed-dose combination, FDC) seems to be an easy and straightforward choice to solve this problem. Unfortunately, in some cases, the difference in physicochemical properties of the combined active pharmaceutical ingredients (APIs) does not allow one-dose compatibility [6,7].

Combining different drugs in a single solid form could potentially offer a solution to this problem, allowing the overcoming of potential issues observed with fixed-dose combinations, and at the same time, allowing for improving physicochemical properties and even bioactivity [8,9,10,11]. To this end, cocrystallization and salt formation are two main approaches in the attempt to combine different drugs into one crystal structure [11]. To date, dozens of drug–drug cocrystals have been reported [4,6,8]; some of them (e.g., meloxicam-aspirin, piracetam–lithium chloride, and curcumin–pyrogallol) exhibit superior solubility profiles compared to the parent compounds [12,13,14]. The cocrystal between monosodium sacubitril and disodium valsartan, which has been marketed by Novartis under the name Entresto for the treatment of chronic heart failure, shows improved bioavailability compared to valsartan [15]. Recently, Hidehiro Uekusa et al. reported [16] a salt–salt antidiabetic drug combination, coupling gliclazide and metformin. This form not only showed improved solubility and dissolution rate characteristics with respect to gliclazide but also solved the hygroscopicity of metformin [16].

The goal of this contribution is to explore a recent design strategy, based on complexation, to combine two APIs into the same solid form. This approach is based on the potential of coupling a second neutral API to a pharmaceutical metal-based salt of the first API, hereby achieving multi-component drug–drug systems. As model systems, we focused on the zinc salts of the carboxylic acid-containing drugs (CADs), ibuprofen, aspirin, and 4-aminobenzoic acid, which we aimed to couple with five pyridine-containing drugs (PDs), nicotinamide, isonicotinamide, isoniazid, amifampridine, and methyl nicotinate, as the nitrogen is expected to easily couple with the metal center. We indeed showed that this approach is highly successful as, in almost all cases, a successful drug–drug solid form was obtained. In addition, zinc ibuprofenate complexes showed improved physicochemical properties compared to the parent compounds.

## 2. Materials and Methods

### 2.1. Materials

Aspirin (ASP), ibuprofen (IBU), nicotinamide (NC), isonicotinamide (INC), isoniazid (INZ), and methyl nicotinate (MN) were bought from Acros (Geel, Belgium). 3,4-diaminopyridine (AMI) was acquired from Sigma-Aldrich (St. Louis, MO, USA), 4-aminobenzoic acid (PABA) from Alfa Aesar (Haverhill, MA, USA), and zinc oxide from Merck (Kenilworth, NJ, USA). All the solvents and reagents were used as received without further treatment.

### 2.2. Preparation of Zinc Ibuprofenate (Zn(IBU)_2_·2H_2_O, Zn(PABA)_2_ and Zinc Aspirinate (Zn(ASP)_2_)

Zinc aspirinate and zinc ibuprofenate were prepared following previously reported experimental procedures [17,18]. Zn(PABA)_2_ was prepared by suspending 4-aminobenzoic acid and zinc acetate in a 2:1 ratio in acetonitrile for 2 days at room temperature. Afterward, the suspension was filtered, and the filtered cake was left to dry under ambient conditions.

### 2.3. Mechanochemical Synthesis of Drug-Drug Complexes

First, 0.25 mmol of zinc oxide, 0.5 mmol of CAD, and 0.25 mmol of PD (Table 1) were added to an Eppendorf tube together with 40 μL of water and 2–3 stainless-steel balls. The mixtures were left to grind at 30 Hz for 90 min using a vibrating mill. Grinding experiments were also performed using double the amount of PD (0.5 mmol) to verify the existence of stoichiometrically diverse complexes. The resulting PXRD patterns were compared to the parent compounds as well as to the grinding results of the binary combination of CAD and zinc oxide to exclude false positives due to the potential formation of the physical mixture between a CAD-Zn salt and neutral PD.

### 2.4. Single Crystal and Bulk Material Preparation

Although the different crystals and bulk materials were obtained in similar procedures, slight variations were applied. Table 2 gives an overview of the ratios and preparation techniques used. 

### 2.5. Powder X-ray Diffraction (PXRD) and Variable-Temperature X-ray Powder Diffraction (VT-PXRD)

Powder X-ray diffraction was conducted on a Siemens D5000 diffractometer (Munich, Germany) equipped with a Cu X-ray source operating at 40 kV and 40 mA (λ = 1.5418 Å) from 4 to 50 degrees at the rate of 0.6 degrees per minute. VT-XRPD profiles were collected on a PANalytical X’Pert PRO automated diffractometer (Malvern Panalytical, Malvern, UK) at a scanning range of 2θ values from 3° to 40°, equipped with an X’Celerator detector and an Anton Paar TTK 450 system (Rigaku, Tokyo, Japan) for measurements at a controlled temperature. Data were collected in the open air in Bragg–Brentano geometry, using Cu–Kα radiation without a monochromator.

### 2.6. Single Crystal Diffraction

Data were collected on a MAR345 image plate detector (marXperts, Norderstedt, Germany) using Mo Kα radiation (λ = 0.71073 Å), generated by an Incoatec IµS microfocus source (Geesthacht, Germany). Data integration and reduction were performed by CrystAlis^PRO^ (Rigaku), and the implemented absorption correction was applied. The structure solution was performed by the dual-space algorithm in SHELXT, and the structure was further refined against F^2^ using SHELXL2014/7 [19]. All non-hydrogen atoms were refined anisotropically, and hydrogen atoms were placed at calculated positions with temperature factors set at 1.2 Ueq of the parent atoms (1.5 Ueq for methyl and OH hydrogens).

### 2.7. Thermogravimetric Analysis (TGA)

TGA analyses of all samples were performed from 30 °C to 500 °C using a heating rate of 10 °C/min with a continuous nitrogen flow of 50 mL/min, on a Mettler Toledo TGA/SDTA851e (Columbus, OH, USA).

### 2.8. Differential Scanning Calorimetry (DSC)

DSC measurements were performed on a TA DSC2500. Samples in aluminum Tzero pans with a punctured lid were heated from 20 °C up to 350 °C using a heating rate of 10 °C/min under a 50 mL/min continuous nitrogen flow.

### 2.9. Solubility Measurement

Solubilities were determined for Ibuprofen, Zn(IBU)_2_(H_2_O)_2_, Zn(IBU)_2_(INC)_2_(H_2_O)_2_, and Zn(IBU)_2_(INZ) at room temperature. To do so, an excess amount of compound was suspended in water at room temperature for two days. The suspension was filtered, and the mass of the clear saturated solution was determined. After evaporation of the solvent, the remaining solid was weighed once again, allowing us to determine the solubility. The original residue was verified by XRPD to confirm no solid transition occurred.

## 3. Results and Discussion

### 3.1. Coupling API through Complexation

Converting acidic drugs into the corresponding metal salts is a widely used approach to improve their solubility and thermal stability [20,21,22]. Low-valent metals such as sodium, potassium, calcium, magnesium, or zinc are the most popular cations used in salt formation as they are pharmaceutically acceptable. The use of these metals often leads to 1:1 or 2:1 organic anion-metal cation complexes [23,24]. These cations, however, often favor high coordination numbers (4, 6, or even higher), with full coordination only achievable using perfectly chelated multidentate API. When such full coordination cannot be achieved by the API, water molecules can fulfil this role, explaining the frequent occurrence of ion-coordinated hydrates among such pharmaceutical salts [25,26]. Interestingly, one is not limited to the use of water, as any neutral organic compound linking to a metal cation could be used to fulfil this purpose. This opens an interesting strategy to design multi-component drug solid forms, as an API can be used as such a second component. We decided to test this strategy on the Zn salts of ibuprofen, aspirin, and 4-aminobenzoic acid. Among all pharmaceutically accepted metals, zinc possesses the most records in the CCDC database. This suggests zinc could be a good potential coordination center for a multi-component complexation. In addition, several contributions have demonstrated that converting acidic drugs such as ibuprofen or aspirin into their zinc salts led to improved bioactivity [17,27]. Zinc is furthermore characterized by low toxicity, with symptoms such as nausea, vomiting, and fatigue only occurring with extremely high zinc intake, making it an ideal metal for pharmaceutical purposes [28].

The first step in achieving multi-component systems implies identifying suitable APIs to couple to these salts. Synthon analysis has been shown to be an efficient approach in the context of cocrystal design. Selecting a suitable coformer based on frequently occurring and reproducible patterns of intermolecular interactions often leads to high success rates in cocrystal screening [29,30,31]. A similar strategy can be applied here as metal ions often show favored coordination modes [32,33]. Going through the CCDC, mixed-ligand zinc complexes are often encountered when one compound contains a carboxylic acid, while the other contains a pyridine function [34,35]. We, therefore, selected a series of five different pyridine-based drugs (nicotinamide, isonicotinamide, isoniazid, amifampridine, and methyl nicotinate) to test the coupling strategy suggested here.

Following our recent success working on saccharinate-Zn-racetam complexes [36], we used a three-component grinding approach to identify potential drug–drug complexes. In this approach, ZnO, the carboxylic acid drug, and the pyridine-containing drug are ground together, leading to a multi-component complex. Table 1 shows that out of the 15 potential combinations tried here, 11 show a different PXRD profile compared to the parent compounds upon grinding (Table 1, Appendix A). To confirm these hits were not due to the mere formation of different polymorphs or that the complexes contained only a single drug component, single crystals of the multi-component complexes were grown. This was successfully achieved for 10 of the complexes. Zn(ASP)_2_-AMI is also expected to be a drug–drug metal complex, but up to now, we have not yet been able to grow single crystals suitable for structural analysis. The four remaining combinations, namely (Zn(IBU)_2_-MN, Zn(ASP)_2_-NC, Zn(ASP)_2_-INZ, and Zn(PABA)_2_-INZ), were not successful in a mechanochemical approach, with an oil-like substance occurring upon grinding. This can be due to a low melting eutectic or a highly hygroscopic product. These combinations, however, can still be achievable if the right conditions are applied (e.g., crystals of complex Zn(PABA)_2_-INZ in the form of methanolate solvate were obtained from solution crystallization).

It should be mentioned that, in most cases, successful cocrystal formation also occurs (SI) between the organic compounds (without the need for the zinc cation). However, as the acids are under ionized form in the complexes, large changes in properties such as solubility can be expected compared to a mere binary cocrystal. Furthermore, complexes form for those cases where the binary combination is unsuccessful.

### 3.2. Structural and Thermal Characterization of Drug-Drug Complexes

Analysis of the 11 crystal structures allows us to regroup them into three different categories according to the observed coordination mode. All three coordination modes (Figure 1) are typically observed for Zn complexes. Zn(PABA)_2_(INC)_2_·0.5H_2_O, Zn(PABA)_2_(NC)_2_, Zn(PABA)_2_(AMI)_2_, Zn(IBU)_2_(INZ), Zn(IBU)_2_(INC)_2_, Zn(PABA)_2_(INZ)·CH_3_OH, and Zn (ASP)_2_(INC)) show a tetrahedral coordination mode. Zn(ASP)_2_(MN), Zn(PABA)(Ac)(MN)·H_2_O, Zn(IBU)_2_(AMI), and Zn(IBU)_2_(NC) show a pyramidal paddle-wheel coordination around the zinc cation. Finally, a hexa-coordinated octahedron mode is observed for the Zn(IBU)_2_(INC)_2_(H_2_O)_2_ and Zn(PABA)_2_(MN)_2_·H_2_O complexes. Zn(IBU)_2_-PDs will be described in more detail below as their examples cover each coordination mode. Furthermore, ibuprofen has low thermal stability (79.04 °C m.p.) and low solubility [37], so we investigated the potential of these complexes to improve drug properties. A mere transformation of ibuprofen into its zinc salt does not solve the issue of low thermal stability as it crystallizes as a dihydrate, which becomes amorphous upon dehydration at essentially the same temperature as the melting point of ibuprofen (see Appendix A).

#### 3.2.1. Tetra-Coordinated Zn(IBU)_2_(INC)_2_ and Hexa-Coordinated Zn(IBU)_2_(INC)_2_(H_2_O)_2_

The tetrahedral coordination mode is the most frequently observed in our results. Zn(IBU)_2_(INC)_2_ shows a central zinc cation tetrahedrally coordinated by two oxygens from two ibuprofen carboxylate groups and two pyridine nitrogen atoms from two isonicotinamide molecules giving an overall Zn(IBU)_2_(INC)_2_ monomer (Figure 2a). Furthermore, the 3D network shows one-dimensional chains formed through NH_amide_···O_carboxylate_ hydrogen bonding between adjacent monomers (Appendix A).

Interestingly, looking for a single crystal, a hydrated complex Zn(IBU)_2_(INC)_2_(H_2_O)_2_ was encountered. It shows a hexacoordinated octahedral coordination mode in which the zinc cation is coordinated by two oxygens from water molecules, two oxygens from ibuprofenate moieties, and two pyridine nitrogens from isonicotinamide (Figure 2b). Unlike for Zn(IBU)_2_(INC)_2_, the neighboring monomers interact via a two-dimensional hydrogen bonding network, which is built by NH_amide_···O_carboxylate_ and OH_water_···O_carboxylate_ interactions (Appendix A). This observation also shows that water molecules have the potential to complete the coordination mode around the central cation and that hydrated phases can also be encountered for drug–drug complexes.

Thermal analysis of Zn(IBU)_2_(INC)_2_ shows a single melting event at 176 °C (Figure 3a). It highlights the remarkable thermal stability of Zn(IBU)_2_(INC)_2_ (isonicotinamide melts at 120 °C [37]). Furthermore, this melting point is also substantially higher than that of Ibuprofen (79.04 °C [36]) and the 1:1 ibuprofen-isonicotinamide cocrystal (119 °C [38]).

The thermal behavior of the hydrated complex—Zn(IBU)_2_(INC)_2_(H_2_O)_2_—shows a 4.75% weight loss at approximately 100 °C corresponding to the loss of two coordinated water molecules (Figure 3d). This water loss leads to the Zn(IBU)_2_(INC)_2_ phase with a melting point of 196 °C (Figure 3c), which is 20 °C above the temperature reported above. VT-PXRD confirms that upon dehydration, the hydrated complex transforms into a different polymorph of the Zn(IBU)_2_(INC)_2_ complex (Appendix A). If drug–drug complexes are considered for the formulation, it is recommended to always investigate polymorphism and hydrate formation.

#### 3.2.2. Penta-Coordinated Zn(IBU)_2_(AMI) and Zn(IBU)_2_(NC)

Both complexes have similar crystal packings. Only the crystal structure of Zn(IBU)_2_(AMI) will be discussed here. Zn(IBU)_2_(AMI) is characterized by a penta-coordinated mode (Figure 4). One monomeric unit shows two zinc cations surrounded by four bridging carboxylates from four ibuprofen molecules. The pyridine nitrogen atoms from amifampridine coordinate to both zinc cations along the Zn-Zn axis. Monomers form a one-dimensional chain through hydrogen bonds between the amide groups and the carboxylates (Appendix A) in a similar manner to the tetra-coordinated Zn(IBU)_2_(INC)_2_ complex. The overall 3D structure should be seen as the packing result of these one-dimensional chains.

Interestingly, mechanochemical screening revealed the existence of a stoichiometrically diverse complex Zn(IBU)_2_(AMI)_2_. Both Zn(IBU)_2_(AMI) and Zn(IBU)_2_(AMI)_2_ complexes can be obtained from solution, slurrying Zn(IBU)_2_ and AMI in isopropanol under different ratios (see experimental part). As for other drug–drug systems (cocrystals or salts), drug–drug complexes also show the potential for stoichiometrical diversity, besides polymorphism and hydrate formation, once more highlighting the importance of a full solid-state screening when considering these forms for formulation.

Thermal analysis shows single endothermal melting peaks at 137 °C and 157 °C for Zn(IBU)_2_(AMI) and Zn(IBU)_2_(AMI)_2_, respectively (see Appendix A). The thermal stability of the solid form complexes Zn(IBU)_2_(AMI) and Zn(IBU)_2_(AMI)_2_ is between those of the parent compounds (the melting point of amifampridine is 218 °C).

The crystal structure of Zn(IBU)_2_(NC) was recently reported by Moura et al. focusing on bioactivity improvement using this complex [39]. The investigation of thermal stability reveals a temperature comparable to that of nicotinamide [40] (128–129 °C) (Appendix A). Furthermore, Zn(IBU)_2_(NC) shows higher thermal stability compared to the ibuprofen-nicotinamide cocrystal, which shows a melting temperature of 96 °C [37,41].

#### 3.2.3. Tetra-Coordinated Zn(IBU)_2_(INZ)

Zn(IBU)_2_(INZ) is an example of an alternative crystal packing. As in the majority of the complexes described in this work, zinc is coordinated to two oxygen atoms from ibuprofenate and to one pyridine-type nitrogen. Contrary to the other complexes, its coordination not only involves pyridine nitrogens. Specifically, coordination also occurs with a hydrazide group (Figure 5a). As coordinating nitrogen atoms of an isoniazid molecule link to different zinc cations, these molecules act as bridging ligands leading to one-dimensional chains (Figure 5b). An identical coordination mode is observed in the Zn(PABA)_2_(INZ)·CH_3_OH complex (see Appendix A).

A single melting peak is observed at 212 °C (Appendix A) highlighting, once again, the improvement in thermal stability compared to both isoniazid (melting at 172 °C [42]) and zinc ibuprofenate.

### 3.3. Solubility Improvement

The water solubility of salts, cocrystals, or multi-component complexes is only directly comparable to the parent compound if the system behaves congruently. Zn(IBU)_2_(INC)_2_(H_2_O)_2_ and Zn(IBU)_2_(INZ) were shown to be congruent in water, and their solubilities were determined gravimetrically. For easy comparison, the solubility is represented in mmol of ibuprofen per liter. Compared to ibuprofen, Zn(IBU)_2_(INC)_2_(H_2_O)_2_ and Zn(IBU)_2_(INZ) show a 17- and 9-fold increase in solubility, respectively, at room temperature. These complexes, however, do show lower solubility compared to the zinc ibuprofenate dihydrate (5.36 mmol/L) (Figure 6).

This lower solubility compared to the parent salt is not a general rule, as reported complexes between L-proline and diclofenac sodium show improved solubility compared to anhydrous diclofenac sodium. Too few examples currently exist in the literature to draw conclusions on the general behavior of the solubility of the complexes vs. the solubility of parent salts [43].

Interestingly, a strong reduction in solubility is observed for isonicotinamide and isoniazid showing a respective solubility of 1.56 mol/L and 1.02 mol/L, showing a strong impact even for the non-ionized compound involved in the complex.

## 4. Conclusions

We explored a recent strategy for dual-drug solid forms, using complexation to achieve our goal. We showed how zinc salts of APIs containing a carboxylic group can be complexed with drug compounds that contain a pyridine moiety. A straightforward three-component mechanochemical screen using the carboxylic acid API, ZnO, and the pyridine-containing API led to the identification of dual-drug solid forms for almost all combinations. Tetra-, Penta-, and Hexa-coordination were encountered around the zinc cation in agreement with literature-based structures. As for any solid-state form, the dual-drug complexes studied in this work also show solvatism and polymorphism, as well as stoichiometric diversity.

The systems studied did show a substantial improvement in thermal stability and solubility compared to ibuprofen. Considering the high success rate, we believe this approach to be a promising technique to increase form diversity and achieve drug–drug solid systems.

## Figures and Tables

**Figure 1 pharmaceutics-14-01960-f001:**
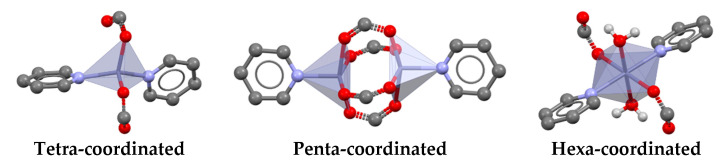
Three favored coordination modes of zinc complexes involving pyridine and carboxylate ligands, as identified in the CSD.

**Figure 2 pharmaceutics-14-01960-f002:**
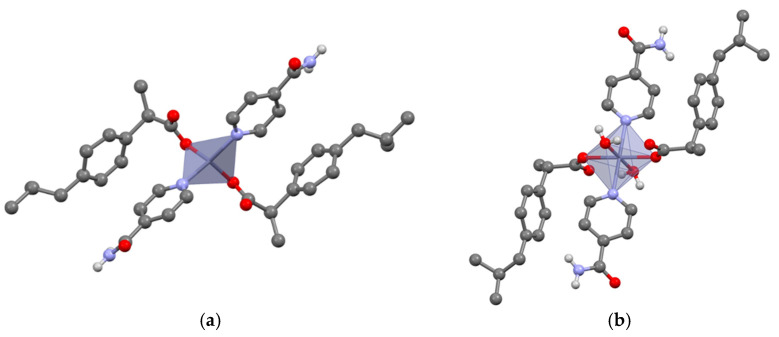
Monomers in Zn(IBU)_2_(INC)_2_ (**a**) and Zn(IBU)_2_(INC)_2_(H_2_O)_2_ (**b**); H_CH_ omitted for clarity.

**Figure 3 pharmaceutics-14-01960-f003:**
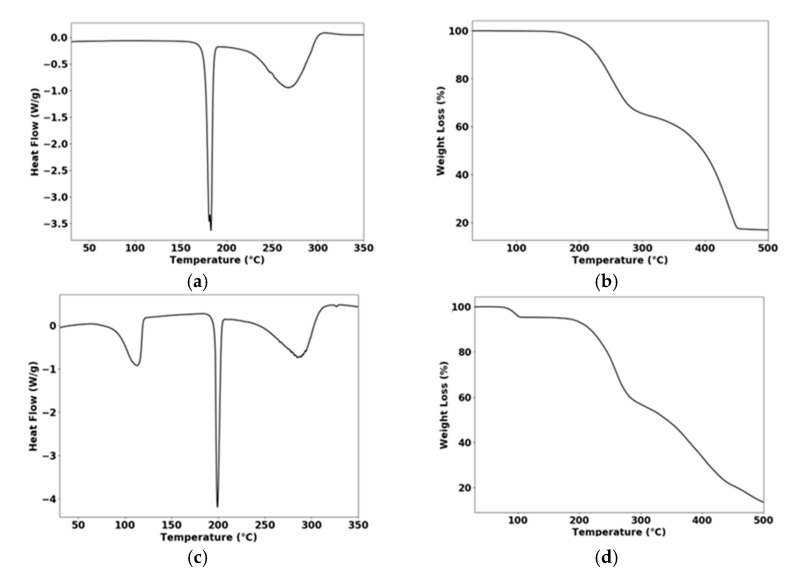
TGA and DSC thermograms of Zn(IBU)_2_(INC)_2_ (**a**,**b**) and of Zn(IBU)_2_(INC)_2_(H_2_O)_2_ (**c**,**d**).

**Figure 4 pharmaceutics-14-01960-f004:**
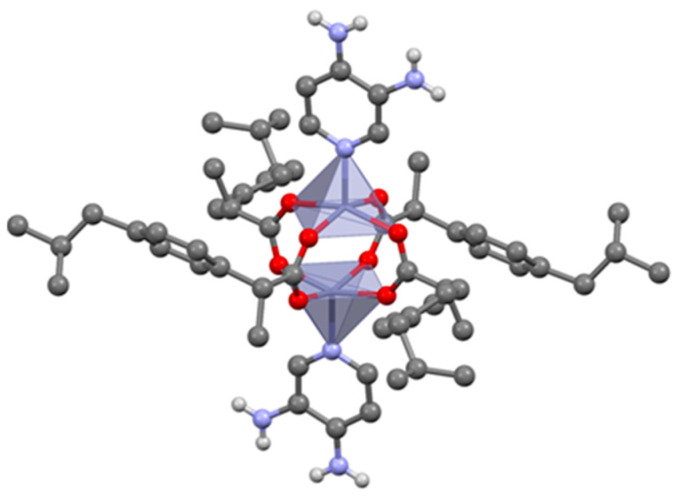
Monomer in Zn(IBU)_2_(AMI), H_CH_ omitted for clarity.

**Figure 5 pharmaceutics-14-01960-f005:**
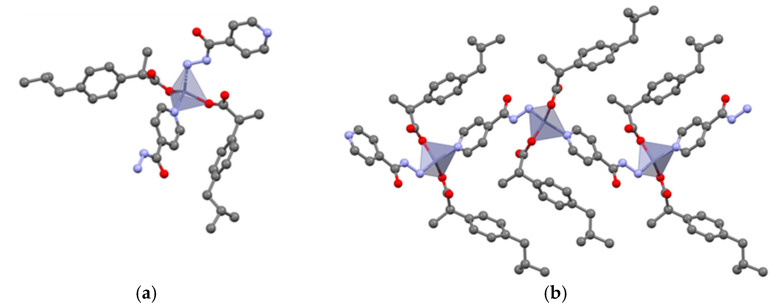
Zn(IBU)_2_(INZ): Tetra-coordinated zinc cation (**a**); one-dimensional chain (**b**). Hydrogen atoms omitted for clarity.

**Figure 6 pharmaceutics-14-01960-f006:**
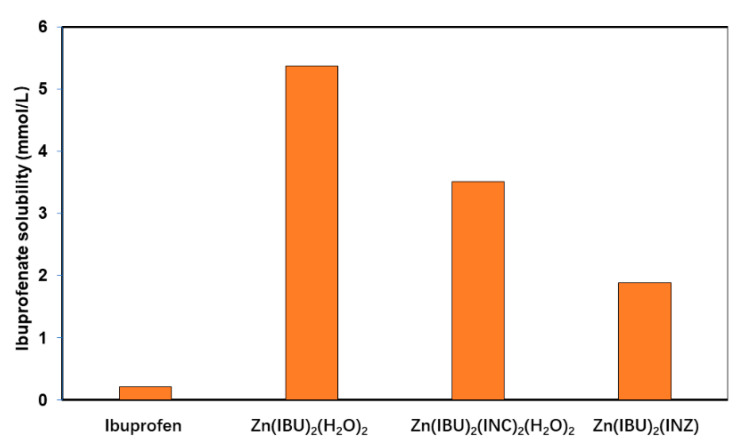
Solubility of ibuprofen, Zn(IBU)_2_(H_2_O)_2_, Zn(IBU)_2_(INC)_2_(H_2_O)_2_, and Zn(IBU)_2_(INZ) calculated based on the amount of ibuprofenate.

**Table 1 pharmaceutics-14-01960-t001:** Complexation screening results. Green stands for the successful mechanochemical combination; grey means an oil-like product was obtained. SC means the combination was confirmed by structural analysis of a single crystal (SC).

	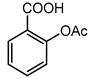 Aspirin (ASP)	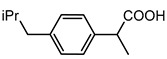 Ibuprofen (IBU)	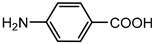 4-aminobenzoic Acid (PABA)
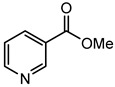 Methylnicotinate (MN)	SC		SC
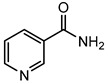 Nicotinamide (NC)		SC	SC
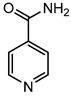 Isonicotinamide (INC)	SC	SC	SC
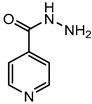 Isoniazid (INZ)		SC	SC ^1^
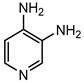 Amifampridine (AMI)		SC	SC

^1^ The complex was obtained in the form of methanolate solvate.

**Table 2 pharmaceutics-14-01960-t002:** Overview of the single crystal and Bulk Material preparation for the reported compounds.

Material	Single Crystal Growth	Bulk Preparation
Zn(IBU)_2_(NC)	Zinc acetate, Ibu, NC dissolved by methanol in 1:2:1 ratio, then evaporation	Slurrying Zn(IBU)_2_ and NC in 1:1 ratio with ethyl acetate as solvent
Zn(PABA)_2_(INC)_2_·0.5H_2_O	Zinc acetate, PABA, INC dissolved by methanol in 1:2:2 ratio, then evaporation	
Zn(PABA)(Ac)(MN)_2_·H_2_O	Zinc acetate, PABA, MN dissolved by methanol in 1:1:2 ratio, then evaporation	
Zn(PABA)_2_(NC)_2_	Zinc acetate, PABA, NC dissolved by methanol in 1:2:2 ratio, then evaporation	
Zn(PABA)_2_(INZ)·CH_3_OH	Zinc acetate, PABA, INZ dissolved by methanol in 1:2:1 ratio, then evaporation	
Zn(IBU)_2_(INC)	Zn(IBU)_2_, INC dissolved by methanol in 1:1 ratio, then evaporation	Slurrying Zn(IBU)_2_ and INC in 1:1 ratio with methanol as solvent
Zn(ASP)_2_(MN)_2_	Zn(ASP)_2_, MN dissolved by methanol in 1:2 ratio, then evaporation	
Zn(ASP)_2_(INC)	Zn(ASP)_2_, INC dissolve by methanol in 1:1 ratio, then evaporation	
Zn(PABA)_2_(MN)_2_·H_2_O	Zn(PABA)_2_, MN dissolve by methanol in 1:2 ratio, then evaporation	
Zn(IBU)_2_(AMI)	Zn(IBU)_2_, AMI dissolved by methanol in 1:4 ratio, then evaporation and cool in fridge	Slurrying Zn(IBU)_2_ and AMI in 1:1 ratio with isopropanol as solvent
Zn(IBU)_2_(INC)_2_(H_2_O)_2_	Zn(IBU)_2_, INC dissolved by methanol/H_2_O mixed solvent (7:3) in 1:2 ratio, then evaporation	Slurrying Zn(IBU)_2_ and INC in 1:2 ratio with water as solvent
Zn(IBU)_2_(INZ)	Zn(IBU)_2_, INZ dissolved by methanol/H_2_O mixed solvent (7:3) in 1:1, then evaporation	Slurrying Zn(IBU)_2_ and INZ in 1:1 ratio with water as solvent
Zn(PABA)_2_(AMI)_2_	stoichiometric quantities of Zinc acetate, Ibu, NC dissolve in methanol than evaporation	

## Data Availability

Crystal data can be obtained free of charge via www.ccdc.cam.ac.uk/conts/retrieving.html (or from the Cambridge Crystallographic Data Centre, 12 Union Road, Cambridge CB21EZ, UK; Fax: +44-1223-336-033; or e-mail: deposit@ccdc.cam.ac.uk). CCDC 2194913–2194925.

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
