# Peer review of "Complexation: An Interesting Pathway for Combining Two APIs at the Solid State"

_pharmaceutics, 2022, doi:10.3390/pharmaceutics14091960_

Round 1

Reviewer 1 Report

The article is well written and has demonstrated the strategy of creating new multi-component solid forms through complexation of zinc salts of carboxylic acid-containing drugs with pyridine-containing drugs. The choice of both classes of materials is well supported and the data is comprehensive. Some of the suggestions to improve the article are provided below.

The experimental methodology provided is a bit confusing to comprehend. If the methodology adopted for various combinations is provided in a table, similar to Table 1 in the manuscript, that might help. Experimental conditions for all solid forms need to be provided so that they are easily reproducible by others too. Also, the list of solvents used for the preparation of bulk materials needs to be provided.

The refcode for each solid form data that has been deposited in CCDC needs to be provided.

Listing/marking the characteristic peak values of new solid forms in their PXRD patterns provided in the supplementary information will be beneficial for other readers.

Author Response

The experimental methodology provided is a bit confusing to comprehend. If the methodology adopted for various combinations is provided in a table, similar to Table 1 in the manuscript, that might help. Experimental conditions for all solid forms need to be provided so that they are easily reproducible by others too. Also, the list of solvents used for the preparation of bulk materials needs to be provided.

Reply: Done

The refcode for each solid form data that has been deposited in CCDC needs to be provided.

Reply:  Done. Numbers are added to the supporting information.

Listing/marking the characteristic peak values of new solid forms in their PXRD patterns provided in the supplementary information will be beneficial for other readers.

Reply: Done

Reviewer 2 Report

This work presented a multi-component crystallization strategy using complexation exemplified by three carboxylic acid-containing compounds and five pyridine-containing compounds. Cocrystal formation is an interesting topic in crystal engineering and pharmaceutics. This work is expected to be of interest to readers of Pharmaceutics. I would recommend publication after the following questions are properly addressed:

1. Can a carboxylic acid-containing compound (such as ibuprofen) and a pyridine-containing compound (such as INC) form a complex without the presence of zinc ions?

2. Whether the incorporated zinc influences the absorption and metabolism of APIs? What are the potential side effects of taking too much zinc from daily administration?

3. In Figure 6, why do complexes Zn(IBU)2(INC)2(H2O)2 and Zn(IBU)2(INZ) have lower solubility than the zinc ibuprofenate dihyrate? Is the lower solubility of complexes a general observation for such crystals (for example, is it true for aspirin and ibuprofen)?

4. In Figure 6, the authors show the effect of complexation on IBU solubility. How does the complexation affect the solubility of the pyridine-containing compound?

Author Response

Can a carboxylic acid-containing compound (such as ibuprofen) and a pyridine-containing compound (such as INC) form a complex without the presence of zinc ions?

Reply: carboxylic acid and pyridine indeed have compatible synthons in cocrystal design. We performed a cocrystal screen through liquid assisted grinding. The reviewer was indeed right that most binary combinations also lead to cocrystal formation. However, this is not the case for eg.  Aspirin-methylnocotinate. Also, our approach often leads to high melting points, which is a supplementary benefit. Following the reviewer, we did mention this finding in the article. 

Aspirin (ASP)

 Ibuprofen (IBU)

4-aminobenzoic acid (PABA)

Methylnicotinate (MN)

liquid

liquid

yes

Nicotinamide (NC)

yes

yes

yes

Isonicotinamide (INC)

yes

yes

yes

Isoniazid (INZ)

yes

no

yes

Amifampridine (AMI)

salt

liquid

yes

Whether the incorporated zinc influences the absorption and metabolism of APIs? What are the potential side effects of taking too much zinc from daily administration?

Reply: Indeed, the introduction of zinc influences the absorption and metabolism of APIs. According to a previous paper (10.1007/s10787-017-0361-0), The transport of organic ligands into the cells can be facilitated by the formation of metal complexes. Therefore, the same therapeutic effect may be achieved with less drug intake.

As for the potential toxicity, G J Fosmire gives a good summary in 1990(10.1093/ajcn/51.2.225). As he said, Zinc is considered to be relatively nontoxic, particularly if taken orally. Symptoms (nausea, vomiting, epigastric pain, lethargy, and fatigue) will occur with extremely high zinc intakes. Considering the most of drugs are only gave in a limited amount and the improved absorption and metabolism efficiency of zinc salt will decrease drug dosage further, side effect of zinc introduction should not be a big problem.

Both references and mention of these effects have been added/are mentioned in the paper now.

In Figure 6, why do complexes Zn(IBU)2(INC)2(H2O)2 and Zn(IBU)2(INZ) have lower solubility than the zinc ibuprofenate dihyrate? Is the lower solubility of complexes a general observation for such crystals (for example, is it true for aspirin and ibuprofen)?

Reply: In fact, the lower solubility of complex than salt is not a general observation. For example,  a previous report demonstrated two complexes between L-proline and diclofenac sodium, both of them possess improved solubility comparing to  anhydrous diclofenac sodium, which is also what we expect in our experiments (10.3390/pharmaceutics12070690). Unfortunately, our experiments prove the solubility improvement is not guaranteed through complexing with a second molecular. Until now, based on very limited example, it is still hard to summarize the rule of solubility modulation after complexation. The reference and some sentences were also added to the paper.

In Figure 6, the authors show the effect of complexation on IBU solubility. How does the complexation affect the solubility of the pyridine-containing compound?

The solubility of isonicotinamide and isoniazid are 1.56mol/L and 1.02mol/L respectively, which clearly are decreased largely because of the combination with zinc ibuprofenate. These values are also added to the paper.

Round 2

Reviewer 2 Report

The authors have addressed questions in the first review round. The manuscript can be published as it is now.